# Gait and Postural Control Deficits in Diabetic Patients with Peripheral Neuropathy Compared to Healthy Controls

**DOI:** 10.3390/bioengineering12101034

**Published:** 2025-09-26

**Authors:** Safi Ullah, Kamran Iqbal, Muhammad Rizwan

**Affiliations:** School of Engineering and Engineering Technology, University of Arkansas at Little Rock, 2801 S University Ave, Little Rock, AR 72204, USA; safi@cytoastra.com (S.U.); mrizwan@ualr.edu (M.R.)

**Keywords:** diabetic peripheral neuropathy, gait analysis, postural stability, balance assessment, fall risk

## Abstract

Diabetic peripheral neuropathy (DPN) is a common complication of type 2 diabetes that impairs gait and balance, increasing fall risk. This study investigated gait characteristics and postural control in individuals with DPN, compared to age- and gender-matched healthy controls. Fifteen DPN patients and fifteen controls underwent assessments of gait, static balance, and mobility. Gait parameters were measured during overground walking using motion capture and force platforms. Static balance was evaluated via tandem stance tests (eyes open/closed), while mobility was assessed with the Timed-Up-and-Go (TUG) test. Dynamic stability was assessed by computing the center-of-pressure Time-to-Contact (TTC) with the mediolateral (ML) stability boundary. We hypothesized that patients with DPN would exhibit an altered gait and reduced ML postural stability during walking. The study results show no significant differences in ML center-of-pressure (COP) excursion or its velocity during walking between groups. Patients with DPN walked relatively slowly, with shorter steps, and showed markedly poorer static balance (earlier failure during tandem stance test), as well as slower TUG performance. Clinically, these findings support routine fall risk screening in DPN using both static balance tests (e.g., tandem stance) and mobility measures (e.g., TUG or gait speed). These findings further suggest that while dynamic postural control during walking may be preserved, DPN patients exhibit gait adaptations and significant static balance deficits, highlighting the need for comprehensive balance assessment in this population.

## 1. Introduction

Diabetic peripheral neuropathy (DPN) is a prevalent neuromuscular disorder that primarily affects patients with type 2 diabetes. Elevated blood glucose levels contribute to nerve damage, leading to reduced sensation, mainly in the lower limbs [1,2,3,4]. Common DPN symptoms include tingling, numbness, burning sensations in the feet, and pain in the lower-extremity muscles. DPN represents a late-stage complication of type 2 diabetes, progressively impacting lower-limb muscles and impairing functional capacity, thereby increasing the risk of falls [5]. Given the high global prevalence of type 2 diabetes and the occurrence of peripheral neuropathy among affected individuals, investigating postural control mechanisms in patients with DPN is crucial for mitigating fall risk.

Balance impairment is noticeable among patients with DPN, resulting in altered gait patterns and movement deficits because of DPN-associated pain and sensory deficits. Compared to healthy individuals, patients with DPN face a significantly higher risk of falls due to postural balance deficits [1,2,3,6,7,8]. Falls pose a considerable health concern, particularly among elderly patients with DPN [5,9,10,11]. Studies have consistently highlighted sensorimotor deficits in patients with DPN, contributing to postural and gait instability [5,12,13]. Reduced sensorimotor function in the lower extremities is strongly associated with increased falls in the elderly population [14]. Patients with DPN are up to 20-fold more likely to fall than age-matched non-diabetic controls [15]. Consequently, elderly DPN patients often alter their pace, showing greater gait variability, which indicates unstable walking [16]. These findings support the perception that patients with DPN may face challenges in maintaining postural control during gait. Investigating gait in patients with DPN may elucidate the mechanisms behind impaired postural control and offer potential for early detection of DPN and associated fall risks in patients with DPN and related neuromuscular disorders.

The assessment of postural control in this study encompassed both static balance during stance and dynamic postural control during gait. Static balance was evaluated via a tandem stance test, while dynamic postural control was assessed during walking using the modified TTC method. Increased COP displacement during standing typically correlates with compromised balance and elevated fall risk. Although previous research has explored postural control in individuals with DPN during quiet and single-leg standing, indicating poorer performance and greater COP excursions compared to healthy controls [17], it remains unclear whether these individuals exhibit altered postural control during dynamic activities such as walking. Therefore, further investigation into postural control under dynamic conditions among patients with DPN may offer valuable insights into lower gait efficiency and fall risks. Given that falls are strongly influenced by mediolateral (side-to-side) control, this study emphasizes mediolateral stability as a central focus of the investigation.

The TTC, also referred to as the time-to-boundary, serves as a measure for assessing postural stability [18,19]. The TTC estimates the time required for the COP to reach the boundary of the foot [18], encompassing both spatial and temporal aspects of postural control, including velocity and acceleration relative to the base-of-support (BOS) [19]. To address challenges in applying standard TTC analysis to dynamic activities like walking, a modified TTC method has been proposed. This modification accounts for the necessity of the COP transitioning from one foot boundary to another as the body moves forward during gait, thus enabling the assessment of postural adjustments that are important for maintaining balance and stability. Previous studies have investigated postural stability in patients with DPN under static conditions, such as quiet standing and single-limb stance, with eyes open and closed [17,20,21]. However, there is a lack of research specifically examining dynamic postural stability, which involves maintaining balance during routine activities such as walking. Another study focused on standard TTC analysis during quiet standing [17]; however, the modified method permits evaluation of postural stability during gait. This modified approach proved effective in assessing postural stability in patients with posterior tibial tendon dysfunction (PTTD) during walking [22]. Therefore, this innovative TTC approach holds great significance for investigating postural control mechanisms in individuals with DPN during walking.

This is the first study to apply a modified TTC approach to gait in individuals with DPN. Prior TTC work in DPN has focused on static conditions (e.g., quiet standing), whereas our approach evaluates stability during dynamic conditions. The modified TTC accounts for the COP’s transition between foot boundaries across single-support phases, enabling quantification of dynamic balance control. Recent applications of TTC to walking in other pathologies support the feasibility of this dynamic assessment framework. The objective of this study was to analyze gait and postural control mechanisms under both static and dynamic conditions in patients with DPN, in comparison to age- and gender-matched healthy controls. This was achieved through a series of trials designed to assess functional mobility and static and dynamic balance; gait parameters (e.g., velocity, step length, stride length, phase durations) were analyzed during walking trials. Static balance was assessed using the tandem stance test (eyes open/closed), dynamic postural control during gait was evaluated with the modified TTC method, and mobility was assessed through the TUG test. Inter-group and intra-group comparisons were performed using statistical analysis. We hypothesized that during the single-support phase of walking, patients with DPN would exhibit a decreased ML TTC percentage, reduced ML COP excursion, and lower ML COP velocity compared to their healthy counterparts. These alterations suggest compromised postural stability mechanisms and are likely to contribute to impaired balance control and decreased functional mobility in the DPN population.

## 2. Methods

### 2.1. Participants

Participants aged between 40 and 75 were recruited from Little Rock Walk-In Clinic in Little Rock, AR, USA, under the supervision of an endocrinologist and medical doctor. Fifteen individuals diagnosed with DPN agreed to participate in the study (see Table 1). Inclusion criteria comprised the following: (1) a minimum 7-year history of type 2 diabetes mellitus; (2) hemoglobin A1c (HbA1c) levels equal to or greater than 7; (3) absence of other comorbidities impacting pain and function; (4) a BMI of less than 35; and (5) no prior history of falls and capability of walking 10 m unaided. Diagnosis of DPN was confirmed by a board-certified physician based on standard clinical criteria, including characteristic neuropathic symptoms (e.g., numbness, tingling, burning in the feet) combined with objective evidence of sensory loss (e.g., abnormal 10-g monofilament test and reduced vibration perception). Study participants with DPN reported engaging in regular light-to-moderate physical activities such as daily walking and playing tennis. The Institutional Review Board (IRB) of the University of Arkansas at Little Rock (UALR) and the University of Arkansas for Medical Sciences (UAMS) approved the study, with all participants providing written informed consent. Enrollment (*n* = 15 per group) was constrained by COVID-19 safety restrictions and IRB-imposed limits on in-person research; therefore, we adopted a literature-consistent sample size appropriate for exploratory analysis of gait and stability. Fifteen healthy controls, matched for age, gender, and BMI, were recruited for comparison with the 15 patients with DPN (see Table 1). Control participants were sourced through flyers, emails, and word of mouth, all recruited between July 2021 and October 2021. Control participants were free of any pathological conditions that hindered unassisted walking.

### 2.2. Experimental Procedures

Three-dimensional kinematic data were collected using 10 CCD cameras (Vicon, Oxford, UK) operated at 100 Hz. A total of six markers were placed on each foot during the walking trials: (1) the medial midfoot, (2) the lateral midfoot, (3) the heel (calcaneus), (4) the toe region (head of the first metatarsal, near the hallux), (5) the medial malleolus, and (6) the lateral malleolus. The heel, toe, medial midfoot, and lateral midfoot markers defined the BOS rectangle used in the TTC calculations (Appendix A). Four in-ground force platforms (AMTI, 40 cm × 60 cm), sampled at 1000 Hz, were used to collect force and COP data (Figure 1). All gait trials were conducted at the UAMS Orthopaedics Biomechanics Research Laboratory. Video and force platform data were synchronized using Vicon Nexus software (Vicon, Oxford, UK). Participants were instructed to (1) perform a tandem balance stance with eyes open and eyes closed for a period of 20 s, and (2) walk on a 10 m walkway at their preferred pace, until three complete data trials were captured. A minimum of three foot contacts within the borders of the four force platforms were required for a trial to be deemed complete. The time-to-failure during tandem balance stance and the time to complete the TUG test were measured using a stopwatch.

## 3. Data Processing

Gait parameters were extracted from the time-series data by identifying distinct events (e.g., heel strike and toe off) and measuring the temporal and spatial intervals between them. Single-support and double-support phases were identified using the vertical ground reaction forces and toe velocities. Toe velocities were utilized to determine gait events for the contralateral limb due to either a limited number of force plates or off-foot contact within the force platform. The double-support ratio was computed as the ratio of double-support time to single-support time. COP excursions and mean COP velocities in both the anterior–posterior (AP) and mediolateral (ML) directions were assessed during the single-support phase for each foot. COP velocities and accelerations were calculated using the first central difference method. The boundaries of the foot (BOS) were defined by the coordinates of the heel, toe, medial midfoot, and lateral midfoot markers (Figure 2), creating rectangular bounds for each foot.

Since the COP shifts between the boundaries of each foot during walking, a modified version of the TTC was utilized [24,25,26]. The ML COP positions, velocities, and accelerations were incorporated into Equation (Equation 1) to compute the ML TTC. COP velocity and acceleration were calculated using the first central difference method.(1)$${\mathrm{TTC}}_{i}=\frac{-v_{i}\pm \sqrt{v_{i}^{2}-2a_{i}d_{i}}}{a_{i}}$$

$d_{i}$: ML distance from COP to ML boundary.

$v_{i}$: ML COP velocity.

$a_{i}$: ML COP acceleration.

i: each data point (100 Hz).

The TTC was calculated at each data point and then compared to the remaining time in the single-support phase. If the TTC was less than the remaining single-support time, the TTC value was stored for that time point, indicating that a postural adjustment was needed. The percentage of TTC was then calculated by dividing the average TTC during single-support by one-half of the single-support time. A percentage of TTC of 100% indicated that no postural adjustment was necessary during the single-support phase. Because the TTC percentage is calculated relative to the duration of the single-support phase and the BOS of the foot in contact with the ground, it reduces dependence on the absolute gait speed compared to the raw excursions or the COP velocities. The mean ML COP trace was determined during the single-support phase and then normalized to the ML boundaries (Figure 2). By normalizing ML COP traces and computing TTC percentages, this novel approach facilitates the identification of balance impairments, providing insights into dynamic balance control mechanisms during walking.

### Statistical Analysis

Seven dependent variables were identified for data analysis: the double-support ratio, AP and ML COP excursions, mean AP and ML COP velocities, and AP and ML TTC percentages. The double-support ratio and COP-based parameters were calculated using a custom MATLAB R2019b (The MathWorks, Inc., Natick, MA, USA) code (MathWorks Inc., Natik, MA, USA). We hypothesized that individuals with DPN would demonstrate a reduced mediolateral TTC percentage, COP excursion, and COP velocity during single-support walking, as well as impaired static balance control, compared to healthy controls. Unpaired t-tests were performed where appropriate using the SPSS statistical package (IBM SPSS Statistics for Windows, Version 23.0. Armonk, NY, USA: IBM Corp.). For comparisons of statistical significance, Cohen’s d effect sizes were calculated and interpreted as follows [27]: 0.20–0.49 = small effect; 0.50–0.79 = medium effect; and ≥0.8 = large effect. The level of statistical significance for all tests was set at *p* < 0.05. Given the exploratory design and sample size (15 per group), we did not include covariates to avoid overfitting; conclusions are framed conservatively.

## 4. Results

Postural control was quantified using several key measures. The excursion of the COP represents the total movement of the COP, reflecting the magnitude of postural sway. COP velocity is the average speed of COP movement, indicating how quickly balance adjustments are made. The percentage of TTC reveals how much of the available single-support time remains before the COP reaches the edge of the BOS, normalized to one-half of the single-support duration. COP trace describes the normalized ML COP path within the foot boundaries during the single-support phase. Together, these measures capture both the magnitude and timing of balance adjustments. During walking, the two groups did not significantly differ in their dynamic postural control metrics (i.e., double-support ratio, COP excursions and velocities, TTC percentages; all *p* > 0.05). Consistently with this observation, group differences in the ML stability outcomes were small in magnitude (Table 2). However, gait and static balance measures differed significantly among groups: patients with DPN walked relatively slowly and with shorter steps, showed substantially poorer tandem stance performance, and exhibited slower TUG times. Static balance was markedly poorer in patients with DPN (earlier loss of balance in tandem stance, especially with eyes closed), and their functional mobility was slower on TUG (Table 3).

In summary, while no significant differences were found in postural control during dynamic walking, significant differences were observed in gait parameters, static balance control, and mobility measures. Patients with DPN exhibited a slower gait velocity, shorter step length, and longer stride duration compared to healthy controls. The static balance control was significantly impaired in patients with DPN under both eyes-open and eyes-closed conditions, accompanied by a significant reduction in the time to failure during the tandem balance stance test. Patients with DPN also demonstrated slower performance during the TUG test compared to healthy controls.

## 5. Discussion

The primary aim of this study was to investigate gait characteristics and postural control in patients with DPN compared to age- and gender-matched healthy controls. We had initially hypothesized that patients with DPN would exhibit significant gait alterations due to dynamic balance control deficits resulting from nerve damage and muscle weakness; however, our findings were more nuanced. We observed no significant differences in key dynamic postural control measures, such as the double-support ratio, COP excursion (both anterior–posterior and medial–lateral), and COP velocity, between patients with DPN and healthy controls. However, despite the absence of postural control differences, patients with DPN demonstrated several gait alterations, including a slower cadence, smaller step length, and longer stride duration.

Because the DPN group walked slower, gait speed represents a potential confound. Our primary dynamic outcome (TTC percentage) is normalized to the single-support time and the BOS, which attenuates trivial speed dependence; nevertheless, residual speed effects cannot be fully excluded. With the present sample size, *n* = 15/15, adding covariates would substantially reduce the degrees of freedom and risk overfitting. Accordingly, we interpret the small, non-significant ML differences cautiously and identify speed-matched or covariate-adjusted analyses as priorities for a larger follow-up study.

The study results suggest that patients with DPN may adopt compensatory strategies in their gait to maintain postural stability, despite the sensory deficits associated with neuropathy. Quantitatively, the lower gait velocity outcome (≈0.9–1.0 m/s) and shorter step length in the case of patients with DPN align with prior reports of reduced speed and step length; similarly, our markedly slower TUG times are consistent with elevated functional mobility impairment reported in neuropathic populations [5,7,9,15]. Likewise, the strong tandem stance deficits mirror prior study findings of impaired static balance in DPN cohorts. To explore this compensatory interpretation, we conducted exploratory within-group correlations (DPN only) between gait velocity and mediolateral stability metrics. Individuals with a slower gait tended to exhibit smaller ML COP excursions and lower ML COP velocity (negative correlations, r≈−0.3 to −0.4; not statistically significant at *n* = 15). Although underpowered, these trends are directionally consistent with a conservative gait strategy that reduces mediolateral challenge to maintain stability.

We should note that our DPN cohort reported regular light-to-moderate physical activity (e.g., daily walking), which likely mitigated gait and balance deficits. This constitutes a selection bias toward higher function and may underestimate impairment relative to more sedentary DPN populations. Sedentary cohorts frequently show a slower gait and poorer balance than observed here, reinforcing the need to interpret our dynamic stability findings in the context of an active DPN sample. Alternative explanations also warrant consideration. First, the modified TTC may be insensitive to subtle intra-group differences when participants adopt cautious gait patterns (slower speed, longer support time), keeping the COP well within foot boundaries. Second, task difficulty may have produced a ceiling effect for dynamic measures, whereas static tasks (tandem stance, eyes closed) exposed deficits. Third, our inclusion of capable walkers with no recent falls likely excluded the most impaired individuals, further attenuating between-group differences in dynamic metrics. The DPN patients in this study were not significantly overweight. Consequently, while their weight may not have contributed substantially to the observed gait differences, it may still have played a role in the subtle compensations observed. Importantly, these patients were physically active, regularly participating in sports and exercise routines, which likely helped to preserve some neuromuscular function and balance, mitigating the more severe gait disorders observed in sedentary DPN patients or individuals with other comorbidities such as obesity. Our findings thus highlight the important role of physical activity and good physique management in maintaining gait function and balance in patients with DPN.

Despite the relatively mild gait alterations, we did observe significant balance deficits in patients with DPN under static conditions. In the tandem balance stance test, which was performed for 20 s with both eyes open and closed, patients with DPN demonstrated significantly poorer balance compared to healthy controls. This finding was particularly pronounced under the eyes-closed condition, when reliance on proprioception is heightened. The absence of significant differences in dynamic postural control may reflect compensatory adaptations during walking, such as a slower gait speed, increased stride duration, and greater reliance on visual input. Static balance tasks, lacking momentum and continuous gait-based sensory updates, may expose deficits that dynamic movement can mask. The deterioration in balance under the eyes-closed condition likely reflects sensory loss severity and the challenges patients with DPN face in relying on their remaining sensory systems (e.g., vestibular input and vision) to maintain stability.

In addition to balance testing, we also assessed mobility using the TUG test. As expected, DPN patients were significantly slower to complete the TUG task compared to healthy controls. This finding aligns with previous studies, which have shown that DPN patients often exhibit slower mobility due to both sensory deficits and reduced muscle strength, making it more difficult to perform tasks that require rapid postural adjustments and coordination [15,28,29]. The slower performance on the TUG test in our cohort further emphasizes the impact of DPN on functional mobility, even in patients who engage in regular physical activity. While they may compensate for some gait deficits during walking, the additional demands of rapid movement and postural control in functional tasks like the TUG test reveal the underlying mobility impairments in this population.

Although our DPN cohort demonstrated compensatory adaptations in their gait, such as a slower gait speed and longer stride duration, the balance deficits observed under static conditions and slower mobility in functional tasks indicate that the neuropathy still exerts a significant impact on overall balance and function. The lack of significant differences in postural control measures like COP excursion and velocity could be explained by the fact that these patients may rely more heavily on compensatory strategies (such as slower movements and greater reliance on visual inputs) to stabilize their COP during dynamic gait tasks. The absence of marked differences in these dynamic measures suggests that while patients with DPN may not exhibit overt postural control deficits, their ability to adjust their posture efficiently during walking is still impaired by the underlying neuropathy.

These findings highlight the complexity of DPN and its effects on movement. While the physical activity levels of our cohort appear to have mitigated some of the more severe gait and postural control deficits typically seen in patients with DPN, their performance on balance and mobility tests clearly shows that they still face challenges. The results suggest that regular physical activity, which likely enhances strength, coordination, and proprioception, plays a critical role in minimizing the functional deficits associated with DPN. However, despite these compensations, patients with DPN remain vulnerable to balance and mobility impairments, particularly in situations requiring quick postural adjustments or when visual feedback is limited.

This study was conducted under COVID-19 constraints; IRB approval and in-person lab access limited recruitment, yielding 15 DPN and 15 control participants. With this sample size (two-sided α=0.05), the study had ∼80% power to detect large effects (Cohen’s d≈0.75−0.80), but was deficient in detecting small-to-moderate effects. Accordingly, non-significant differences in dynamic postural control should be interpreted with caution, given the risk of Type 2 error. To aid interpretation, we report effect sizes alongside *p*-values in Table 2 and Table 3. Additionally, while we controlled for age and gender, we did not match for other potential confounders, such as specific comorbidities or detailed measures of neuropathy severity. Further studies with larger, more diverse cohorts, including individuals with varying levels of DPN severity and different activity levels, are needed to better understand the full impact of physical activity on gait and balance in patients with DPN.

## 6. Conclusions

Patients with DPN demonstrated compensatory gait adaptations and maintained similar postural control patterns to healthy controls in dynamic conditions, while they exhibited significant balance deficits in static conditions and slower mobility during functional tasks. Clinically, our results support routine fall risk screening in DPN, combining static balance tests (e.g., tandem stance, ideally with eyes closed) and dynamic mobility measures (e.g., TUG, stride length, or gait speed). Even in physically active patients with DPN with near-normal dynamic stability, static balance testing and functional mobility testing can reveal hidden deficits, highlighting the need for targeted balance and gait rehabilitation. Future research should explore interventions that could further improve balance and mobility in patients with DPN, with a particular focus on enhancing proprioceptive feedback and postural control strategies.

## Figures and Tables

**Figure 1 bioengineering-12-01034-f001:**
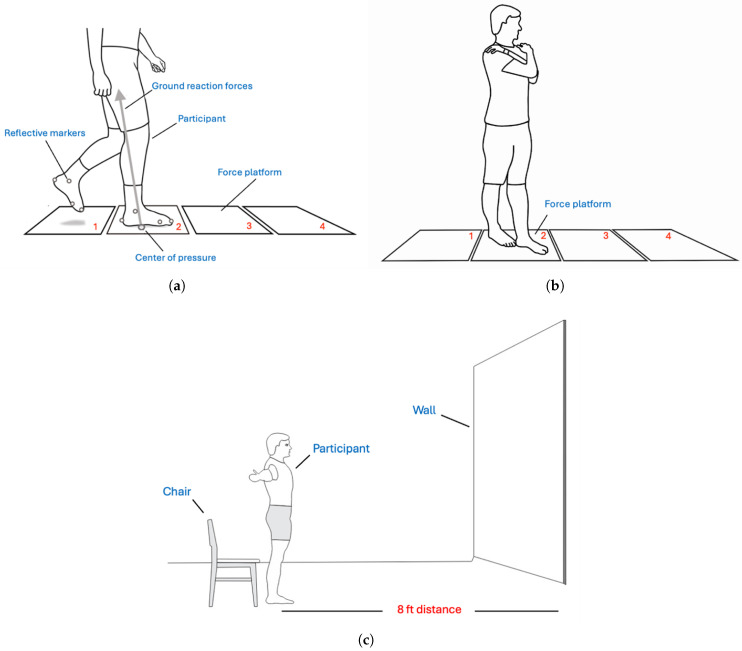
Experimental setup for balance tests. (**a**) Gait analysis setup with four in-ground force platforms to record center-of-pressure (COP) movement during the single-support phase of walking. (**b**) Tandem stance test (eyes open and eyes closed) to assess static balance. (**c**) Timed-Up-and-Go (TUG) test setup for functional mobility assessment.

**Figure 2 bioengineering-12-01034-f002:**
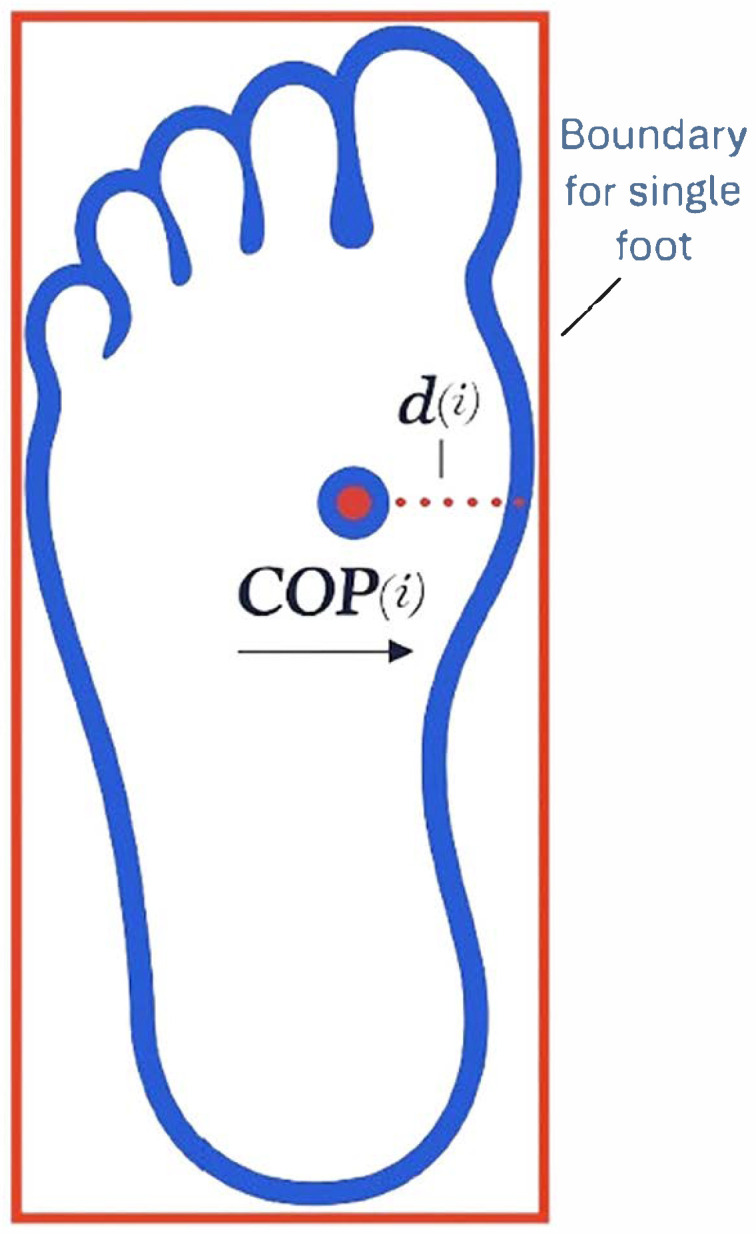
Illustration of the rectangular boundary of the foot; foot confines (BOS) are defined by heel, toe, medial-midfoot, and lateral-midfoot markers (see Section 2) [23].

**Table 1 bioengineering-12-01034-t001:** Participant demographics.

Parameter	DPN Group (*n* = 15)	Control Group (*n* = 15)
Gender (M/F)	11/4	11/4
Age (years)	61.5 ± 7.9720	59.13 ± 7.85
Height (m)	1.70 ± 0.08	1.75 ± 0.09
Weight (kg)	83.66 ± 12.21	81.54 ± 20.42
BMI (kg/m^2^)	28.92±4.46	26.31±4.66
DM Type	Type 2	NA
HbA1c	8.2 ± 1.4	NA
Duration of Diabetes	14.33 ± 4.98	NA

Note: Groups did not differ in age, height, or BMI (all *p* > 0.05, independent-samples *t*-tests), and values are presented as mean ± SD. **DPN:** diabetic peripheral neuropathy, **DM:** diabetes mellitus, **BMI:** body mass index, **HbA1c:** glycated hemoglobin test.

**Table 2 bioengineering-12-01034-t002:** Comparison of postural control parameters between DPN and healthy control groups.

Variables	DPN (*n* = 15)	Control (*n* = 15)	*p*-Value	Effect Size (D)
Double-support ratio (%)	33.6 ± 5.3	31.8 ± 6.1	0.523	0.3
AP COP excursion (cm)	13.5 ± 2.8	14.3 ± 2.1	0.541	0.3
ML COP excursion (cm)	1.9 ± 0.8	1.7 ± 0.9	0.687	0.2
AP COP velocity (cm/s)	30.4 ± 6.2	32.2 ± 4.4	0.381	0.3
ML COP velocity (cm/s)	6.9 ± 2.7	7.0 ± 3.2	0.917	0.0
AP TTC percentage (%)	75.2 ± 8.6	73.8 ± 9.5	0.655	0.2
ML TTC percentage (%)	83.2 ± 7.5	81.0 ± 6.8	0.442	0.2
ML COP trace (%)	50.1 ± 9.4	51.2 ± 10.1	0.724	0.1

**Table 3 bioengineering-12-01034-t003:** Results for gait parameters, balance, and mobility.

Variables	DPN (*n* = 15)	Control (*n* = 15)	*p*-Value	Effect Size (D) **
Gait velocity (m/s)	1.02 ± 0.09	1.38 ± 0.04	<0.001 *	3.9
Step length (m)	0.65 ± 0.18	0.73 ± 0.12	0.035 *	0.8
Stride length (m)	0.83 ± 0.08	1.26 ± 0.10	<0.001 *	5.3
Single-support duration (s)	0.65 ± 0.13	0.49 ± 0.11	<0.001 *	1.3
Stride duration (s)	1.54 ± 0.10	1.15 ± 0.11	<0.001 *	4.1
Tandem balance (Eyes open, s)	9.21 ± 4.64	20.0 ± 0.0	0.008 *	1.7
Tandem balance (Eyes closed, s)	4.27 ± 3.69	18.26 ± 2.25	0.004 *	1.8
Timed Up and Go (TUG), (s)	17.81 ± 1.74	11.45 ± 1.16	0.001 *	2.2

* *p* < 0.05, ** Cohen’s d effect size.

## Data Availability

The data presented in this study are available from the principal author upon reasonable request. Data are not publicly available due to privacy and ethical restrictions.

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
