# Peer review of "Gait and Postural Control Deficits in Diabetic Patients with Peripheral Neuropathy Compared to Healthy Controls"

_bioengineering, 2025, doi:10.3390/bioengineering12101034_

Round 1

Reviewer 1 Report

Comments and Suggestions for Authors

The manuscript addresses an important clinical problem: gait and balance impairments in patients with diabetic peripheral neuropathy (DPN). The study is well-motivated, and the methodology is described with reasonable clarity. The use of motion capture and force platforms is appropriate, and the findings add nuance to our understanding by distinguishing between preserved dynamic postural control and impaired static balance.

That said, the manuscript would benefit from a clearer articulation of the novelty of its contribution, more rigorous handling of limitations, and stronger integration with the existing literature. The discussion sometimes reiterates results without sufficiently expanding on their implications. Finally, the manuscript would be improved by careful editing for conciseness and consistency.

Major Comments

The manuscript needs to clarify what is novel about applying the modified Time-to-Contact (TTC) method to DPN patients. This is introduced as innovative, but the discussion does not fully explain how this approach advances the field compared to previous studies that have already examined TTC in neuropathic or gait-impaired populations.

With only 15 participants per group, the statistical power is limited. This is acknowledged briefly, but the discussion should more explicitly address the possibility of Type II error, particularly given the negative findings for dynamic postural control. A power analysis (even retrospective) would strengthen the credibility of the conclusions.

The DPN cohort is described as relatively physically active. This is an important confounder that could bias the results toward milder impairment. The discussion should highlight this limitation more directly and consider stratifying or at least qualitatively contrasting the findings with more sedentary DPN populations described in the literature.

The lack of significant group differences in dynamic postural control measures is a central result. However, the discussion leans heavily on compensatory strategies as an explanation without providing supporting data. Were there within-group correlations between gait speed and COP measures? Including such analyses could support the compensatory strategy hypothesis.

Several systematic reviews and recent studies are cited, but the manuscript would benefit from a sharper positioning of the current findings within that body of work. For example: how do these results compare quantitatively with reported gait velocity or tandem stance times in other DPN cohorts? This would help readers appreciate the relative severity of deficits in the current sample.

The conclusion should go further in discussing the translational implications. How should clinicians incorporate these findings into fall-risk screening? Should static balance tests (such as tandem stance with eyes closed) be prioritized in routine evaluation? This would enhance the practical impact of the work.

Minor Comments

The abstract is overly detailed in methodology but light on the significance of findings. Consider emphasizing the clinical implication of static balance deficits rather than reporting exact percentage differences in gait variables.

The epidemiological data on diabetes and DPN (lines 26–33) is useful but could be condensed; the focus should remain on the rationale for studying postural control specifically.

The description of marker placement and force plate use is clear but somewhat technical for a general bioengineering audience. Consider moving some of these details to supplementary material. Please clarify whether all gait trials were conducted at the same facility (UAMS vs UALR) since differences in force platform configuration could influence data comparability.

Tables are clear, but the text repeats much of the numerical detail unnecessarily. Summarize in words and let the tables carry the specifics.

The discussion would be strengthened by a more balanced tone. At present, the “compensatory strategies” explanation dominates, but alternative interpretations (e.g., insufficient sensitivity of measures, ceiling effects, or cohort selection bias) deserve consideration.

Figure legends could be expanded to be self-explanatory (e.g., specify what COP stands for in Figure 1 without requiring the reader to look back).

Comments on the Quality of English Language

Minor grammatical corrections are needed (e.g., “time up and go test” should consistently be “Timed Up and Go test”). Ensure consistency in reporting percentages: sometimes “-26%” is used, elsewhere differences are given in raw values. Standardize. 

Reviewer 2 Report

Comments and Suggestions for Authors
  • It is somewhat awkward that mediolateral (ML) stability is not mentioned in the introduction, yet it suddenly becomes the primary focus of investigation in the hypothesis. Please address this inconsistency.

  • Please specify the diagnostic criteria used for identifying diabetic peripheral neuropathy 

  • Consider adding a figure to illustrate the marker placement.

  • Please provide justification for the chosen sample size.

  • The foot was defined using four markers (heel, toe, medial midfoot, lateral midfoot), which means the forefoot was not modeled. While the Please explain and justify this decision.

  • The current description of the TTC calculation is unclear. Including a figure or flowchart would improve clarity.

  • The difference in walking speed between the two groups is a major concern. Please analyze whether the significant difference in walking speed contributed to the non-significant results of the stability parameters by including walking speed as a covariate.

  • Table 1: Please clarify the notation “DMType (kg).”

Reviewer 3 Report

Comments and Suggestions for Authors

see attached:

Comments:

This study compared the patients with DPN with the healthy using balance data, especially TOC. There are major concerns to be clariid as below. 

  1. Lines 173-176: if p >0.05, there is not significant differences between two groups. Therefore, these statements are not appropriate. Please rewording.
  2. Table 1: is there any p values for age, height and body mass/BMI?
  3. Table 2:is the method to calculate TTC novel or not?
  4. Table 2: are the numbers within brackets standard deviations? Please give a note.
  5. Table2: last 3 rows need to be clarified. What is definition for %, e.g. a partial duration to total duration in a gait cycle? See points 9-10 as well.
  6. Table 3: are the xx in mean+-xx standard deviations? Please give a note.
  7. Figure 2: how to identify the edges of the foot? Clarify this point in Methods please.
  8. Line 53: what is CoP (pressure)? Did you use pressure plate or force platform? If the latter used, this should be CoF (force). Clarify this point.
  9. Line 52: what is the definition of TTC? Clarify this point. From figure 2, the foot boundary is a fixed/dynamic edge of foot printing and CoP is the trace of centre of force. Between these two are distances which keep dynamically changing. The CoP cannot reach the edges of medial and lateral foot-printing. How to estimate the time or duration for the CoP to move to the edges? Did you use the anterior- and posterior-edges in foot-printing? Is TTC from heel-strike to toe-off? Clarify this point.
  10. Line 64: how to determine supporting areas or edges? Clarify this point, maybe relate to the 7).

Reviewer 4 Report

Comments and Suggestions for Authors

The manuscript addresses an important clinical problem by examining gait and postural control in diabetic patients with peripheral neuropathy (DPN). While the topic is relevant and falls within the scope of Bioengineering, the current manuscript suffers from significant scientific and methodological shortcomings that undermine its suitability for publication.

First, the study is based on a very small sample size (15 DPN patients vs. 15 healthy controls), which substantially limits the statistical power and generalizability of the findings. Although this limitation is briefly acknowledged in the discussion, it is not adequately addressed in terms of how it might bias the results. The lack of robust statistical differences in key postural control parameters may simply be due to underpowered analyses rather than meaningful clinical conclusions. Furthermore, the inclusion and exclusion criteria appear inconsistent, and no clear assessment of neuropathy severity (e.g., clinical scales, electrophysiological tests) was reported, making it difficult to evaluate whether the sample truly represents the DPN population.

Second, the novelty of the work is questionable. The manuscript primarily replicates known findings that DPN patients walk slower, with shorter step length, and have poorer static balance. These outcomes are already extensively documented in prior literature. The paper’s supposed contribution—the use of a modified time-to-contact (TTC) method during gait—fails to produce any significant differences, leaving the main hypothesis unsupported. This further diminishes the impact of the work, as the results do not advance our understanding of gait or balance impairments beyond what is already established.

Finally, the manuscript has issues in presentation and interpretation. The discussion stretches the limited findings into broad clinical implications without sufficient evidence, and the overall writing lacks critical engagement with existing literature. Figures and tables present only very basic analyses, and no deeper biomechanical or neuromuscular insights are provided. The conclusion overstates the significance of results that are, in reality, non-contributory.

Round 2

Reviewer 1 Report

Comments and Suggestions for Authors

My concerns have been satisfactorily addressed, and the paper is now acceptable.

Reviewer 2 Report

Comments and Suggestions for Authors

I do not have any additional comments.

Reviewer 3 Report

Comments and Suggestions for Authors

ok now.

Reviewer 4 Report

Comments and Suggestions for Authors

Regardless of narrative trims, the scientific original contribution of the study remains unclear., because methods are classical and no clear scientific novelty appear in system, not even application. While fully respecting the carried out work, I believe that the manuscript at hand does not reach the threshold for further consideration.